# Strain engineering of two-dimensional multilayered heterostructures for beyond-lithium-based rechargeable batteries

Pan Xiong[1,5], Fan Zhang[1,5], Xiuyun Zhang[2,5], Shijian Wang[1], Hao Liu[1], Bing Sun[1], Jinqiang Zhang [1], Yi Sun[2], Renzhi Ma[3], Yoshio Bando[3], Cuifeng Zhou[4], Zongwen Liu [4], Takayoshi Sasaki [3✉] & Guoxiu Wang [1✉]

Beyond-lithium-ion batteries are promising candidates for high-energy-density, low-cost and large-scale energy storage applications. However, the main challenge lies in the development of suitable electrode materials. Here, we demonstrate a new type of zero-strain cathode for reversible intercalation of beyond-$Li^+$ ions ($Na^+$, $K^+$, $Zn^{2+}$, $Al^{3+}$) through interface strain engineering of a 2D multilayered $VOPO_4$-graphene heterostructure. In-situ characterization and theoretical calculations reveal a reversible intercalation mechanism of cations in the 2D multilayered heterostructure with a negligible volume change. When applied as cathodes in $K^+$-ion batteries, we achieve a high specific capacity of 160 mA h $g^{-1}$ and a large energy density of ~570 W h $kg^{-1}$, presenting the best reported performance to date. Moreover, the as-prepared 2D multilayered heterostructure can also be extended as cathodes for high-performance $Na^+$, $Zn^{2+}$, and $Al^{3+}$-ion batteries. This work heralds a promising strategy to utilize strain engineering of 2D materials for advanced energy storage applications.

[1] Centre for Clean Energy Technology, School of Mathematical and Physical Sciences, University of Technology, Sydney, NSW 2007, Australia. [2] College of Physical Science and Technology, Yangzhou University, 225002 Yangzhou, China. [3] International Center for Materials Nanoarchitectonics (WPI-MANA), National Institute for Materials Science (NIMS), Namiki 1-1, Tsukuba, Ibaraki 305-0044, Japan. [4] School of Chemical and Biomolecular Engineering, The University of Sydney, Sydney, NSW 2006, Australia. [5] These authors contributed equally: Pan Xiong, Fan Zhang, Xiuyun Zhang. ✉email: sasaki.takayoshi@nims.go.jp; guoxiu.wang@uts.edu.au

The rapid development of renewable energy resources has triggered tremendous demands in large-scale, cost-efficient and high-energy-density stationary energy storage systems[1]. In this regards, beyond-lithium-ion batteries (LIBs) are recently extensively investigated, including sodium-ion batteries (SIBs), potassium-ion batteries (PIBs), zinc-ion batteries (ZIBs), and aluminum-ion batteries (AIBs)[2–5]. Na, K, Zn, and Al are abundant metallic elements in the earth crust, far exceeding Li (Supplementary Fig. 1a). Similar to Li-based chemistry, the Na and K have suitable redox potentials, endowing SIBs and PIBs with high terminal voltages and potentially high energy densities (Supplementary Fig. 1b). In particular, the ZIBs and AIBs are attractive for their superior theoretical volumetric energy densities because of the multi-electron transfer reactions (Supplementary Fig. 1c). Despite these promising aspects, the development of these beyond-LIBs has been impeded by the lack of suitable electrode materials[5–7].

Layered materials, based on an intercalation mechanism, have been particularly studied in alkali metal-ion batteries for their stable cyclability and high rate capability, benefitting from effective and simple intercalation chemistry of ions into their large interlayer galleries[8]. The first layered electrode materials used in LIBs was transition metal disulfides, such as $TiS_2$, developed by M. Stanley Whittingham in 1976[9,10]. Later, in 1980, John Goodenough and co-workers reported a layered transition metal oxide, $LiCoO_2$, and variants of which are still being used in the majority of smart phone batteries today[11]. Since then, the layered materials have received major research interests as the intercalation electrodes for rechargeable batteries, including beyond-LIBs. Among the intercalation electrodes, zero-strain electrode materials with excellent long cycling performance have attracted great attention due to their negligible lattice parameter change (<1%) during guest ion insertion and extraction. This merit is crucial to meet the long-term cycling requirement of beyond-LIBs, however, has rarely been achieved thus far, to the best of our knowledge[12–14]. Compared with $Li^+$ (0.76 Å), $Na^+$ (1.02 Å) and $K^+$ (1.38 Å) have much larger ionic radii (Supplementary Fig. 1d). Divalent $Zn^{2+}$ and trivalent $Al^{3+}$ typically exhibit stronger electrostatic/Coulombic interactions with the host lattices than the monovalent ions. All these factors have restricted the reversible insertion and diffusion of ions into host lattices and induced huge volume expansion of the electrode materials, leading to a sluggish rate capability and short cycle life.

Two-dimensional (2D) molecular nanosheets produced from exfoliation of their layered precursors have been well recognized in electrochemical energy storage. Through the recently developed exfoliation and assembly strategy, vertical assembly of different nanosheets on top of each other generates 2D multilayered heterostructures as promising layered materials for energy storage[15–18]. Several typical examples have been demonstrated recently for Li and Na storage, including phosphorene/graphene, $MnO_2$/graphene, $Ti_{0.87}O_2$/N-doped graphene and $MoS_2$/graphene[19–22]. Considering the large interlayer galleries between adjacent nanosheets are theoretically applicable for intercalation of various metal ions, it is of great interest to investigate the 2D multilayered heterostructures for beyond-Li ions such as $K^+$, $Zn^{2+}$, and $Al^{3+}$. However, in layered structures, an expansion perpendicular to the layers may induce phase change and even structural collapse upon ion intercalation on the host lattice. This is much more serious for beyond-LIBs based on bulky and multivalent metal ions. 2D heterostructures yield unusual properties and phenomena, benefiting from the synergistic properties of different 2D materials via high-quality heterointerfaces[15,16,23]. Strain engineering at the interfaces is an efficient approach to control the properties of 2D materials, which has been widely demonstrated in electronics and catalysis[24–26]. Only recently, an observation was reported that interface strain engineering of 2D carbon-$MoS_2$ heterostructured nanosheets could control the electrochemical reactivity of $MoS_2$ for Li insertion[27]. Theoretical calculations indicated the possibility of tunable Li storage of 2D transition metal carbides via strain engineering[28,29]. Based on these pioneering results, it is expected the interface strain of 2D heterostructures could accommodate the intercalation of these beyond-$Li^+$ ions for a highly stable cycle life.

Here, we report that interface strain engineering of 2D multilayered heterostructure produces a zero-strain cathode for reversible intercalation of beyond-$Li^+$ ions such as $Na^+$, $K^+$, $Zn^{2+}$, $Al^{3+}$. A reversible intercalation mechanism with a negligible volume change during charge and discharge processes has been demonstrated. Consequently, high-performance SIBs, PIBs, ZIBs, and AIBs are obtained in terms of high specific capacity, large energy density, and long-term cycling stability.

## Results

**Synthesis and characterizations.** A 2D $VOPO_4$-graphene multilayered heterostructure was utilized as a proof-of-concept layered material for our design (Fig. 1). $VOPO_4$ nanosheets were first synthesized by intercalation and exfoliation of bulk layered $VOPO_4·2H_2O$ crystals (Supplementary Fig. 2). Then, the 2D $VOPO_4$-graphene multilayered heterostructures were obtained through a self-assembly process of the $VOPO_4$ and graphene nanosheets (Supplementary Fig. 3). For comparison, the $VOPO_4$ nanosheets were freeze-dried, and nanoflakes of restacked $VOPO_4$ nanosheets were also prepared. When applied as cathodes in beyond-LIBs, the restacked $VOPO_4$ nanoflakes gradually collapsed after cycling due to the severe volume change. In contrast, the 2D multilayered $VOPO_4$-graphene heterostructures could enable long-term cycling stability for reversible intercalation of various cations ($Na^+$, $K^+$, $Zn^{2+}$, and $Al^{3+}$).

The bulk $VOPO_4·2H_2O$ crystals are platelet chunks composed of tightly stacked layers (Fig. 2a and Supplementary Fig. 4). After treated with acrylamide solution, the layered platy morphology was retained (Supplementary Fig. 5). An increased basal spacing of 0.91 nm, larger than that of $VOPO_4·2H_2O$ (0.74 nm), suggests the intercalation of acrylamide (Supplementary Fig. 6). Consequently, the $VOPO_4$ nanosheets were obtained by exfoliation of the intercalated $VOPO_4$-acrylamide. The intercalation-exfoliation protocol produced uniform $VOPO_4$ nanosheets with several hundreds of nanometers in size and a thickness of ~4 nm (Supplementary Fig. 7), corresponding to ~6 monolayers of $VOPO_4$. Transmission electron microscopy (TEM) observation revealed an ultrathin sheet-like morphology of the obtained $VOPO_4$ nanosheets (Fig. 2b and Supplementary Fig. 8). Zeta-potential measurements showed a negative value of −(45 ± 8) mV for the $VOPO_4$ nanosheets, which should be ascribed to the reduction of $V^{5+}$ during the exfoliation process (Supplementary Fig. 9)[30]. The surface charges enabled a stable dispersion of $VOPO_4$ nanosheets with a noticeable Tyndall light scattering (inset of Fig. 2b). Although no surfactant was added during the exfoliation process, the suspension was highly stable for several months.

In a conventional procedure, the nanosheet suspension was freeze-dried to produce lamellar nanoflakes of self-restacked $VOPO_4$ nanosheets (Fig. 2c). Although the thickness is much thinner than the bulk chunks, the restacked nanoflakes still suffer from massively decreased active surfaces. In this work, a 2D multilayered heterostructure was rationally designed by confining the $VOPO_4$ nanosheets between graphene layers. The graphene was modified with a cationic polymer, poly (diallyldimethylammonium chloride) (PDDA), resulting in a positively charged nature[20–22,31]. Due to electrostatic attraction, these two oppositely

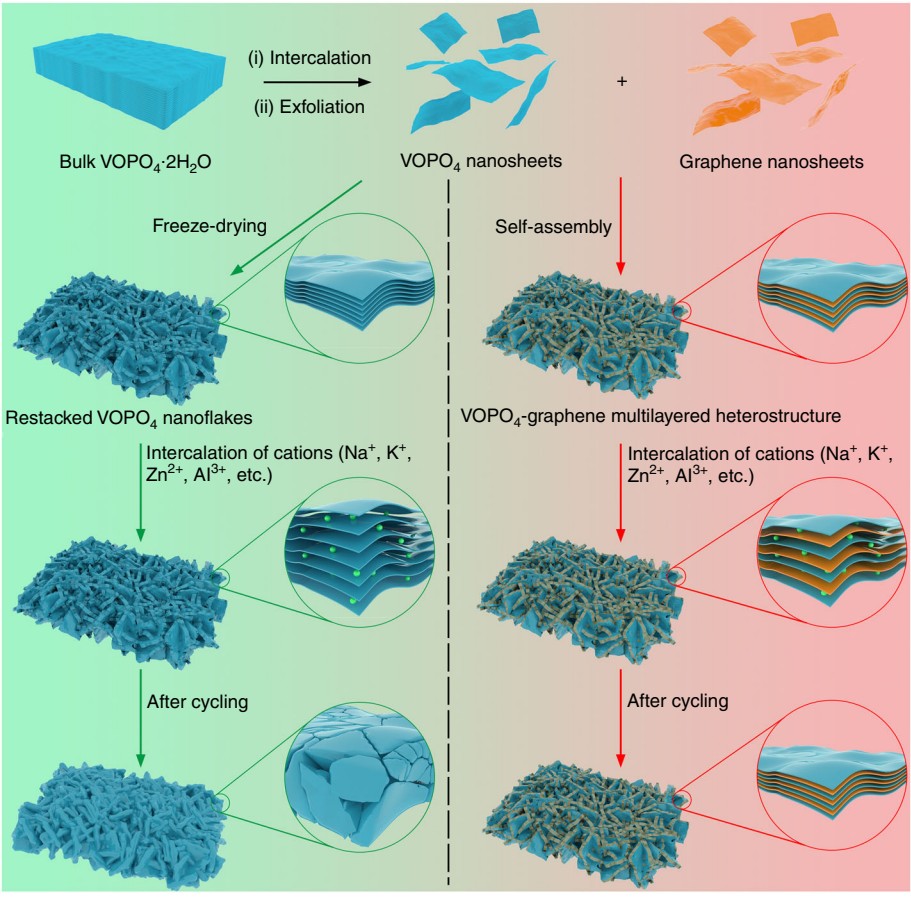

**Fig. 1 2D multilayered heterostructures for intercalation of beyond-Li$^+$ ions.** The 2D multilayered heterostructure is prepared by self-assembly between VOPO$_4$ nanosheets and graphene. However, freeze-drying of suspensions of VOPO$_4$ nanosheets produces lamellar restacked nanoflakes. As electrode materials for intercalation of Na$^+$, K$^+$, Zn$^{2+}$, and Al$^{3+}$ cations, the restacked nanoflakes show poor cycling performance due to the severe volume change and gradually collapsed structure after several cycles. In contrast, the 2D multilayered VOPO$_4$-graphene heterostructure could effectively alleviate the large volume expansion and maintain superior structural stability during the repeated intercalation/deintercalation of various cations, thus enabling long-term capability.

charged nanosheets self-assembled into a face-to-face stacked 2D multilayered heterostructure. An optimized mass ratio between the VOPO$_4$ nanosheets and modified graphene could be theoretically estimated as ~11.4 based on a hypothesized area-matching model (Supplementary Fig. 10). A flocculation was induced immediately after mixing the suspensions of these two nanosheets in this ratio (Supplementary Fig. 3a). The flocculated product was collected, and the content of graphene was estimated to be ~10 wt% (Supplementary Fig. 11). In a control experiment, a stable suspension was obtained after mixing suspensions of the VOPO$_4$ and graphene oxide, both of which are negatively charged nanosheets (Supplementary Fig. 3b). This clearly indicates that the VOPO$_4$ and graphene nanosheets were held together by electrostatic attraction in the resulting 2D multilayered heterostructure. The 2D multilayered VOPO$_4$-graphene heterostructure show a 3D porous structure composed of crumpled thin layers (Fig. 2d, e). A side-view SEM image (Supplementary Fig. 12) indicates a multilayered heterostructure of VOPO$_4$-graphene. The HRTEM image (Fig. 2f) of the cross-section further reveals that the thin layers are multilayered structures of stacked VOPO$_4$ and graphene sheets. The high-angle annular dark-field scanning transmission electron microscope (HAADF-STEM) image and the corresponding elemental mapping indicate the uniform distribution of VOPO$_4$ and graphene nanosheets (Fig. 2g). The selected-area electron diffraction (SAED) pattern exhibits the in-plane diffraction rings of both VOPO$_4$ and graphene nanosheets

(Fig. 2h). These clearly demonstrate that the VOPO$_4$ and graphene nanosheets were assembled into a multilayered heterostructure with a layer-by-layer pattern.

Due to the mismatch of the in-plane lattice spacing between VOPO$_4$ and graphene, a significant interface strain was possibly propagated into the VOPO$_4$ nanosheets from the VOPO$_4$-graphene interfaces. Recent studies showed that in-plane strain can be probed through Raman spectroscopy[32,33]. Raman spectroscopy analysis (Fig. 2i) of the 2D VOPO$_4$-graphene multilayered heterostructure showed the identical signatures of VOPO$_4$ together with the D-band and G-band of graphene. To assess the strain in the 2D VOPO$_4$-graphene multilayered heterostructure, statistical Raman spectroscopy mapping (Fig. 2j, k) comprising over 100 individual scans was performed on the symmetric O–P–O and V=O stretching modes of VOPO$_4$ (910–1050 cm$^{-1}$). After exfoliation, red shifts of Raman bands of these two stretching modes were observed for the VOPO$_4$ nanosheets. However, clear blue shifts of ~15 cm$^{-1}$ for O–P–O and ~3 cm$^{-1}$ for V=O were observed when the VOPO$_4$ nanosheets were stacked with graphene, supporting the presence of interface-induced compressive strain on the VOPO$_4$ sheets. Furthermore, the strain ($\delta$) can be approximately calculated by

$$\Delta\omega = \delta \times \omega/2.66 \qquad (1)$$

in which $\omega$ is the Raman shift and $\Delta\omega$ is the change of Raman peak[34]. Taking the O–P–O model as an example, a strain of

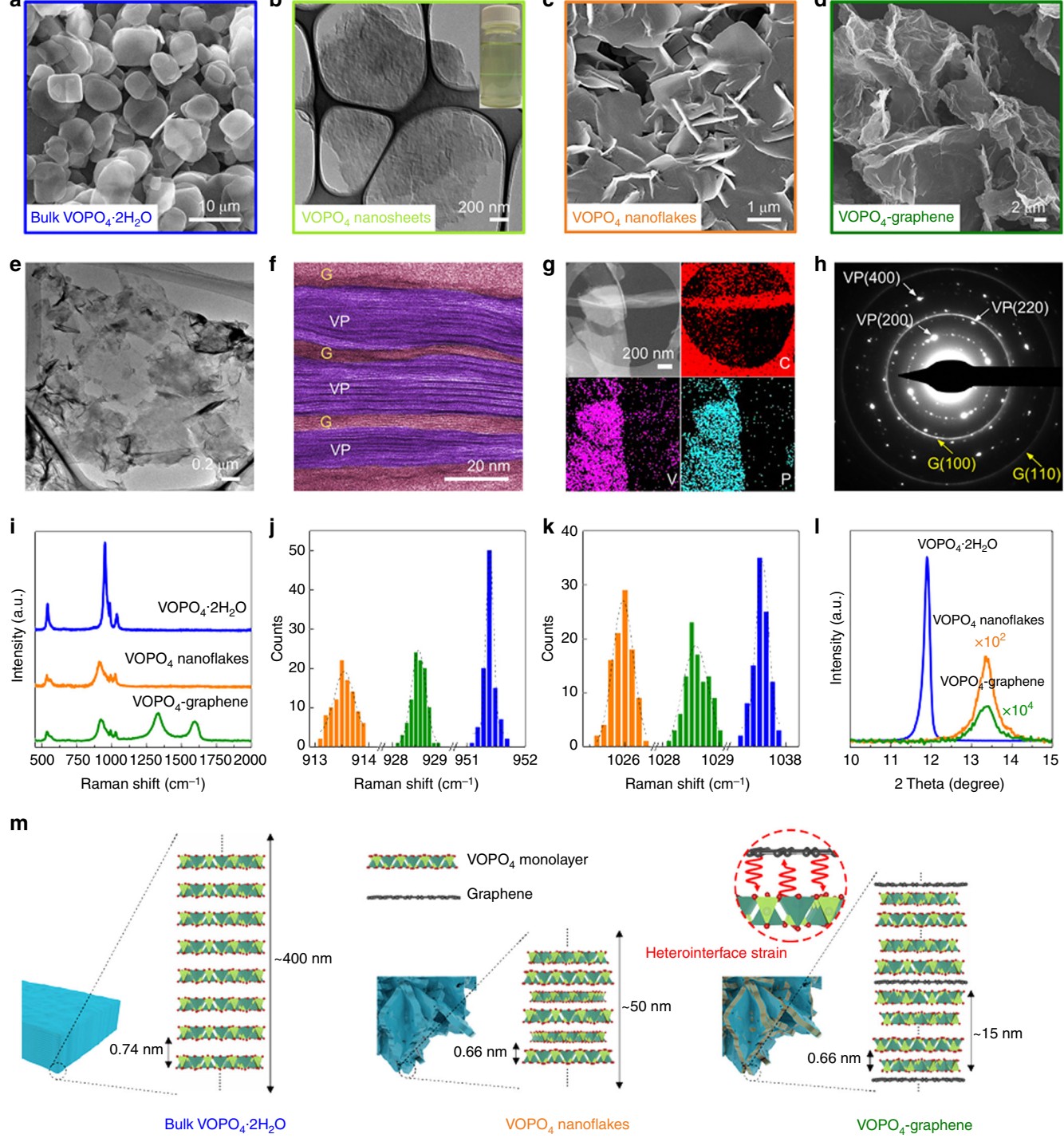

**Fig. 2 Characterizations of 2D VOPO$_4$-graphene multilayered heterostructures. a** SEM image of bulk layered VOPO$_4$·2H$_2$O crystals. **b** TEM image of exfoliated VOPO$_4$ nanosheets. The inset shows the Tyndall light-scattering effect in suspensions of exfoliated nanosheets. **c** SEM image of restacked VOPO$_4$ nanoflakes. **d** SEM and **e** TEM images of VOPO$_4$-graphene. **f** Cross-section HRTEM image of VOPO$_4$-graphene showing the multilayered structure of alternately restacked VOPO$_4$ (VP) and modified graphene (G) nanosheets. **g** HAADF-STEM image and corresponding elemental mapping of VOPO$_4$-graphene. **h** SAED pattern of VOPO$_4$-graphene showing the in-plane reflections of both VOPO$_4$ (VP) and modified graphene (G) nanosheets. **i** Raman spectra of VOPO$_4$·2H$_2$O, VOPO$_4$ nanoflakes and VOPO$_4$-graphene. **j, k** Distribution from 100 individual scans showing average shifts due to interface strain of VOPO$_4$-graphene heterostructures. **l** Comparison of 001 XRD peaks of VOPO$_4$·2H$_2$O, VOPO$_4$ nanoflakes and VOPO$_4$-graphene. The peak intensities of VOPO$_4$ nanoflakes and VOPO$_4$-graphene was multiplied by 10$^2$ and 10$^4$ times, respectively. **m** Schematic illustration of the different restacking and interlayer distances in bulk VOPO$_4$·2H$_2$O, VOPO$_4$ nanoflakes and VOPO$_4$-graphene.

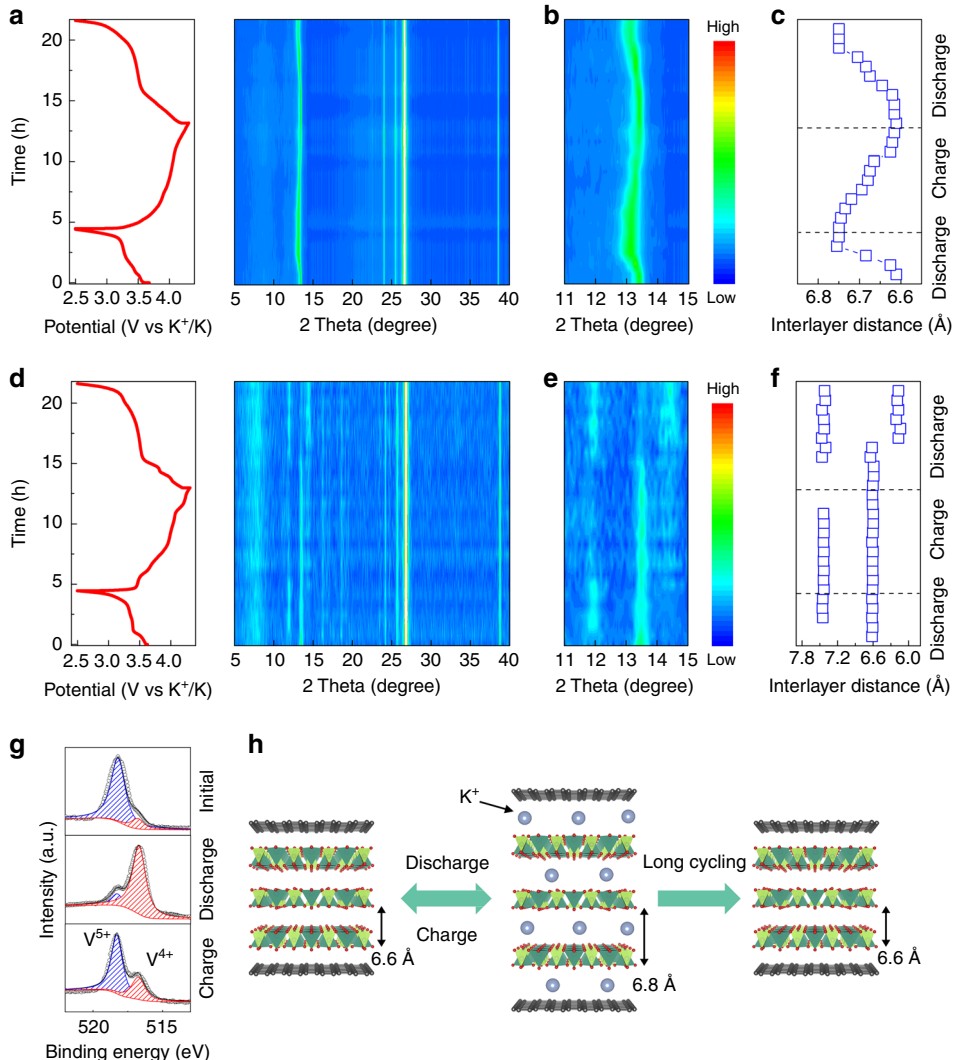

**Fig. 3 Zero-strain intercalation of K$^+$ ions.** Typical charge/discharge profiles and in-situ XRD patterns of **a** VOPO$_4$-graphene and **d** VOPO$_4$ nanoflakes as cathodes for PIBs. Enlarged in-situ XRD patterns from 11° to 15° of **b** VOPO$_4$-graphene and **e** VOPO$_4$ nanoflakes. The interlayer distance of **c** VOPO$_4$-graphene and **f** VOPO$_4$ nanoflakes calculated from XRD patterns during the charge/discharge processes of PIBs. **g** Ex-situ V 2p XPS of VOPO$_4$-graphene in initial, discharged and charged states. **h** Schematic illustration of proposed reversible intercalation mechanism of K$^+$ ions for the 2D VOPO$_4$-graphene multilayered heterostructure.

~4.0% was determined in the 2D VOPO$_4$-graphene multilayered heterostructures. Density functional theory (DFT) calculations based on VOPO$_4$–VOPO$_4$ and VOPO$_4$-graphene bilayers were performed to investigate the lattice strains induced by the interface in the heterostructures. As shown in Supplementary Fig. 13, we observed shortened P–O and V–O bond lengths from 1.597 and 1.550 Å in VOPO$_4$–VOPO$_4$ bilayers to 1.594 and 1.548 Å in VOPO$_4$-graphene bilayers. The lattices (a × b) of VOPO$_4$ were optimized to be 6.212 × 6.212 Å. The lattice strain ($\delta$) for VOPO$_4$-graphene is defined as

$$\delta = \frac{a_{VOPO4-graphene} - a_{VOPO4}}{a_{VOPO4}} \quad (2)$$

A lattice strain ($\delta$) of 3.2% can be estimated in VOPO$_4$-graphene, which is consistent with the result based on the Raman analysis.

The XRD patterns of the VOPO$_4$·2H$_2$O, VOPO$_4$ nanoflakes and 2D VOPO$_4$-graphene multilayered heterostructure are shown in Fig. 2l and Supplementary Fig. 14. All three materials are layered VOPO$_4$ structures with dominant 00l peaks. The bulk VOPO$_4$·2H$_2$O showed an interlayer distance of ~0.74 nm. The shifted 001 peaks resulted in a decreased interlayer separation of

~0.66 nm for the VOPO$_4$ nanoflakes and the 2D VOPO$_4$-graphene multilayered heterostructure. The significantly different intensities of 001 peaks imply the stacking density of VOPO$_4$ in these layered structures, which can be roughly estimated based on the Scherrer formula. Consequently, the 2D VOPO$_4$-graphene multilayered heterostructures exhibited an average restacking thickness of ~15 nm (Fig. 2m), which is much smaller than that of bulk VOPO$_4$·2H$_2$O (~400 nm) and VOPO$_4$ nanoflakes (~50 nm). These results strongly confirm that limited number of VOPO$_4$ sheets were confined and stabilized between the graphene nanosheets, resulting in a 2D VOPO$_4$-graphene multilayered heterostructure with interface strain.

**Zero-strain intercalation mechanism.** Taking potassium as an example, the intercalation mechanism of the 2D VOPO$_4$-graphene multilayered heterostructure was first investigated. In situ XRD measurements of the 2D VOPO$_4$-graphene multilayered heterostructure cathodes were conducted during charge/discharge cycles (Supplementary Fig. 15 and Fig. 3a). A large irreversible capacity was possibly due to the protons in the structure, as elucidated by the previous studies[35–37]. No new peaks or

asymmetric variations were observed, indicating a topotactic one-phase reaction in the 2D VOPO$_4$-graphene multilayered heterostructure cathode during cycling. The 001 peak was observed during the charge/discharge processes, demonstrating the well-maintained 2D multilayered heterostructure. However, it gradually shifted to lower $2\theta$ values upon discharge (potassium intercalation) and reversed back to the original value after charge (potassium de-intercalation), which can be attributed to the continuous lattice volume variations during cycling (Fig. 3b). Interlayer distances were calculated from the 001 peaks during the charge/discharge cycles with reversible lattice breathing (Fig. 3c). The corresponding volume change was calculated to be only 2.0%, comparable to the reported zero-strain electrode materials for PIBs[13,38]. Such a small volume change in the crystal structure guarantees a long-term cycling stability.

For comparison, the structural evolution of the restacked VOPO$_4$ nanoflakes upon K$^+$-ion insertion and extraction was also examined (Supplementary Fig. 16 and Fig. 3d). In addition to the peak shifting, asymmetric peak evolution was clearly identified upon K$^+$-ion insertion and extraction, suggesting a two-phase reaction (Fig. 3e). The initial 001 peak was still detected after the discharge process, indicating an insufficient insertion of K$^+$ ions within the nanoflakes. The evolution of the interlayer distance of restacked VOPO$_4$ nanoflakes is shown in Fig. 3f. A new peak with an increased interlayer distance of ~0.75 nm was observed after K$^+$-ion insertion, resulting in a volume change of 136%, which is much larger than that of 2D VOPO$_4$-graphene multilayered heterostructure. Moreover, during the second cycle of K$^+$-ion insertion, the 001 peak gradually disappeared and another new peak with a decreased interlayer distance of ~0.62 nm was formed, which may imply the degradation of the initial layered structure of restacked VOPO$_4$ nanoflakes. In order to confirm this hypothesis, the restacked VOPO$_4$ nanoflakes and 2D VOPO$_4$-graphene multilayered heterostructure were further charged/discharged for 50 cycles. As shown in Supplementary Fig. 17, the charge/discharge profiles of the 2D VOPO$_4$-graphene multilayered heterostructure overlapped at different cycles. However, the polarization of restacked VOPO$_4$ nanoflakes significantly increased and the discharge plateau vanished after the initial several cycles, accompanied by obviously decayed capacity (Supplementary Fig. 18). After cycling, the 001 peak of the 2D VOPO$_4$-graphene multilayered heterostructure was still maintained, indicating the outstanding structural stability (Supplementary Fig. 19). In contrast, no 001 peak was observed for the restacked VOPO$_4$ nanoflakes. These results prove the collapse of the layered structure of restacked VOPO$_4$ nanoflakes during the repeated intercalation/de-intercalation of K$^+$ ions (Supplementary Fig. 20). In contrast, the 2D multilayered heterostructures with VOPO$_4$ nanosheets confined between graphene layers exhibited a high stability. The reversible reduction and oxidation of V$^{5+/4+}$ upon K-ion intercalation and de-intercalation were also observed (Fig. 3g). After the 1st discharge, the signal of V$^{5+}$ in the XPS spectra was remarkably decreased, confirming the reduction of V during the insertion of K$^+$ ions. After a full charge, a major contribution of V$^{5+}$ and a small amount of V$^{4+}$ were observed. However, the slightly increased quantity of V$^{4+}$ state suggested the charged sample was not exactly recovered to the initial state. This should be ascribed to a slightly irreversible process during the 1st charge/discharge process. Due to a high specific surface area of the multilayered heterostructure, it was reasonable that the partial of the intercalated K$^+$ ions were still trapped in the VOPO$_4$-graphene after the first charge cycle. Correspondingly, we further checked the K 2p signals during the 1st charge/discharge cycle. As shown in Supplementary Fig. 21, the K 2p signals were highly dominated in the discharge process, as evidence for K$^+$ ion intercalation.

However, in the charge process, a slight amount of K was still observed, implying the incomplete extraction of K$^+$ ions.

Unlike the previously reported bulk layered materials[37,39–42], the 2D VOPO$_4$-graphene multilayered heterostructure showed increased interlayer spacing upon K$^+$ ion intercalation. It should be noted that the reported structures were bulk layered compounds with water or metal ions in the interlayer space. Upon discharge, K$^+$ ions were intercalated whereas the crystal water was extracted simultaneously from the structure. Owing to an attractive force between the inserted K$^+$ ions and the lattice oxygen atoms of stacked layers, a decreased interlayer spacing was obtained. However, in our work, the 2D VOPO$_4$-multilayered heterostructure was based on exfoliated VOPO$_4$ nanosheets, in which no intercalated molecular H$_2$O was preserved and the interlayer distance was decreased. This 2D multilayered heterostructure is different from the reported bulk materials. Thus, upon insertion of large K$^+$ ions, a dominant effect of ion intercalation on the host lattice is an expansion perpendicular to the layers[43,44], which induces the expansion of interlayer distance of VOPO$_4$ nanosheets. Here, in the 2D multilayered heterostructure, the VOPO$_4$ and graphene nanosheets were held together by electrostatic attraction. An interface-induced compressive strain on the VOPO$_4$ layers was formed due to the mismatch of the in-plane lattice spacing between VOPO$_4$ and graphene. The interface strain between VOPO$_4$ and graphene is possible to accommodate the expansion for a superior stable intercalation reaction. The reversible change in interlayer distance of VOPO$_4$-graphene should be attributed to the unique multilayered heterostructure and the interface strain between the adjacent VOPO$_4$ and graphene sheets. On the basis of these results, a reversible intercalation mechanism of K$^+$ ions for the zero-strain 2D multilayered heterostructure cathode is proposed and illustrated in Fig. 3h.

DFT calculations were performed to further understand the storage process of K$^+$ ions in the 2D VOPO$_4$-graphene multilayered heterostructure. For an approximate calculation, the restacked bilayer and four-layer structures were used to illustrate the effects of the restacking number of VOPO$_4$ layers on the K storage. We investigated three optimized models of K atoms stored on the (001) and (010) facets of VOPO$_4$, as well as K intercalated into the (001) interlayer of the layered VOPO$_4$ structures (Fig. 4a, b). The calculated binding energies ($\Delta E$) of the three models for the restacked bilayer and four-layer structures were shown in Fig. 4c. It is concluded that K atoms prefer to adsorb on the exposed (010) facets and intercalate into the (001) interlayers rather than adsorb on the exposed (001) facets. Moreover, the binding stabilities of the K storage on the restacked bilayer structure are more stable than that on the four-layer structure, proving the superiority of atomically thin nanosheets in the 2D multilayered heterostructures compared to the restacked nanoflakes. We further investigated the K atom diffusion in the interlayer of restacked VOPO$_4$ bilayers (Fig. 4d) and VOPO$_4$-graphene multilayers (Fig. 4e). As shown in Fig. 4f, the K diffusion in the interlayer of adjacent VOPO$_4$ layers (Path 1 in Fig. 4e) generated a slightly larger energy barrier than that of the interlayer between VOPO$_4$ and graphene (Path 2 in Fig. 4e). It is notable that much lower diffusion energy barriers for K atom diffusion in the VOPO$_4$-graphene multilayers were obtained, compared with that in VOPO$_4$ bilayers (Fig. 4e). This theoretically confirmed the ultrafast K storage capability of the 2D multilayered heterostructures.

**Electrochemical performances**. The electrochemical performance of the 2D VOPO$_4$-graphene multilayered heterostructures was investigated as cathodes for PIBs. Figure 5a shows the

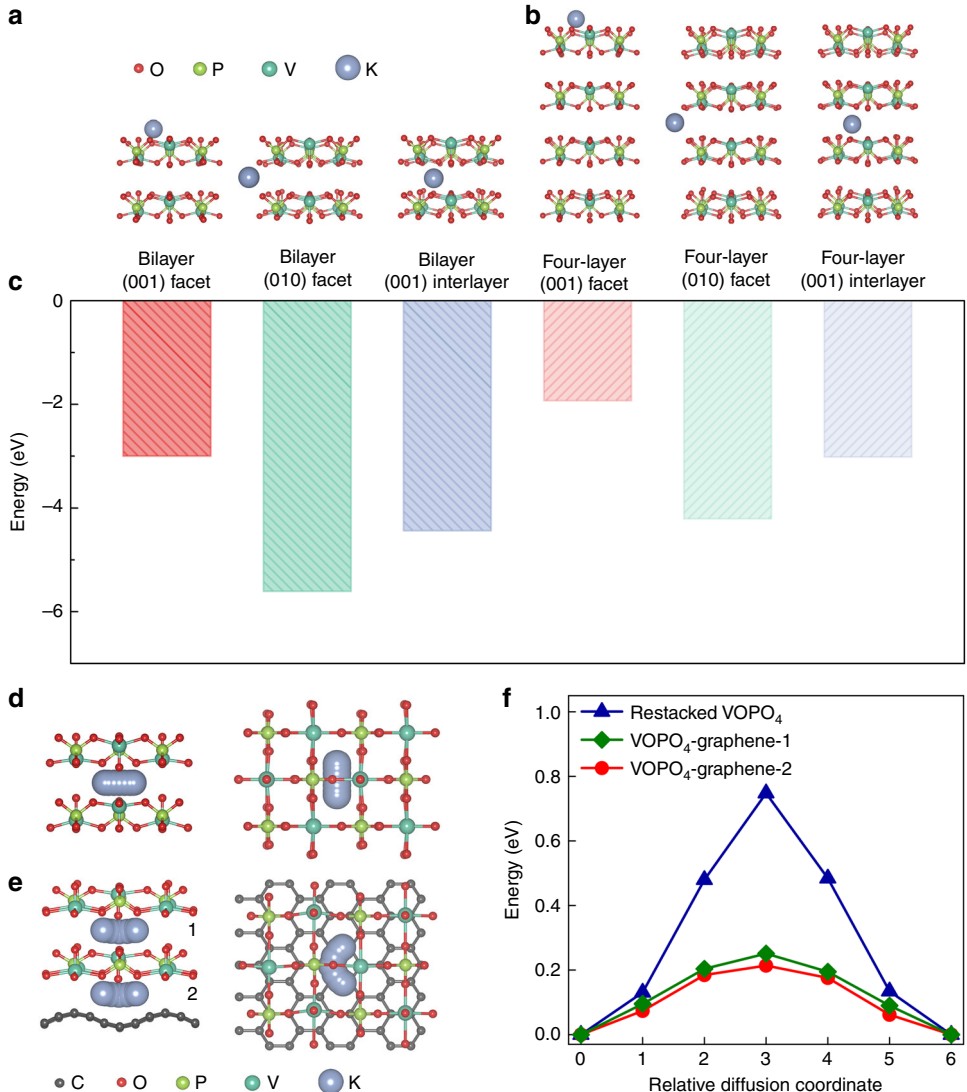

**Fig. 4 Theoretical simulations of K storage and diffusion.** Typical models of K atoms intercalated/adsorbed into/on (**a**) the (001) facet, (010) facet and (001) interlayer of bilayer $VOPO_4$ and (**b**) the (001) facet, (010) facet and (001) interlayer of four-layer $VOPO_4$. **c** Corresponding binding energies of the six models in a and b. Typical models of K atom diffusion path in the interlayer of (**d**) restacked $VOPO_4$ and (**e**) $VOPO_4$-graphene. **f** K diffusion energy profiles of restacked $VOPO_4$ and $VOPO_4$-graphene.

galvanostatic charge/discharge voltage profiles of the as-prepared layered cathodes. Distinct charge and discharge plateaus were observed corresponding to the reduction and oxidation of $V^{5+/4+}$ during $K^+$-ion insertion and extraction. The 2D $VOPO_4$-graphene multilayered heterostructure showed an average charge/discharge voltage of ~3.5 V versus $K^+/K$. The lowest polarization among the three samples suggests the improved kinetics of $K^+$ ion diffusion in the 2D multilayered heterostructure. A reversible specific capacity of ~165 mA h $g^{-1}$ was delivered by the 2D $VOPO_4$-graphene multilayered heterostructure cathode, which is higher than that of bulk $VOPO_4·2H_2O$ (~100 mA h $g^{-1}$) and restacked $VOPO_4$ nanoflakes (~160 mA h $g^{-1}$). In a control experiment, practically no capacity was exhibited for the modified graphene under the same condition (Supplementary Fig. 22). Upon continuous charge/discharge, the 2D $VOPO_4$-graphene multilayered heterostructure delivered a stable reversible capacity of ~160 mA h $g^{-1}$ without obvious capacity decay during 100 cycles (Fig. 5b). In contrast, the bulk $VOPO_4·2H_2O$ suffered a sharp decay of capacity by 30 cycles. Although the restacked $VOPO_4$ nanoflakes exhibited improved cycling performance than

the bulk crystals, the specific capacity still obviously decreased from 160 to 100 mA h $g^{-1}$ in less than 50 cycles.

The 2D $VOPO_4$-graphene multilayered heterostructure also exhibited superior rate capability compared with the bulk $VOPO_4·2H_2O$ and restacked $VOPO_4$ nanoflakes, especially under high current densities (Fig. 5c). High reversible capacities of ~160, 150, 130, 120, 110, 100, and 90 mA h $g^{-1}$ were delivered at 0.1, 0.2, 0.5, 1, 2, 5, and 10C, respectively. Even at a high current density of 20C, a stable reversible capacity of ~80 mA h $g^{-1}$ was still maintained. This value is approximately four times higher than that of the restacked $VOPO_4$ nanoflakes. The corresponding charge/discharge profiles of the 2D $VOPO_4$-graphene multilayered heterostructure cathode were shown in Supplementary Fig. 23. All the profiles showed the same shape with gradually increased shifting and overpotential, indicating high reversibility. Furthermore, high-rate cycling performance was also measured for the 2D $VOPO_4$-graphene multilayered heterostructure cathode. A symmetric K–K cell was also fabricated and consistently showed stable K stripping/plating polarizations at different current densities (Supplementary Fig. 24). Figure 5d

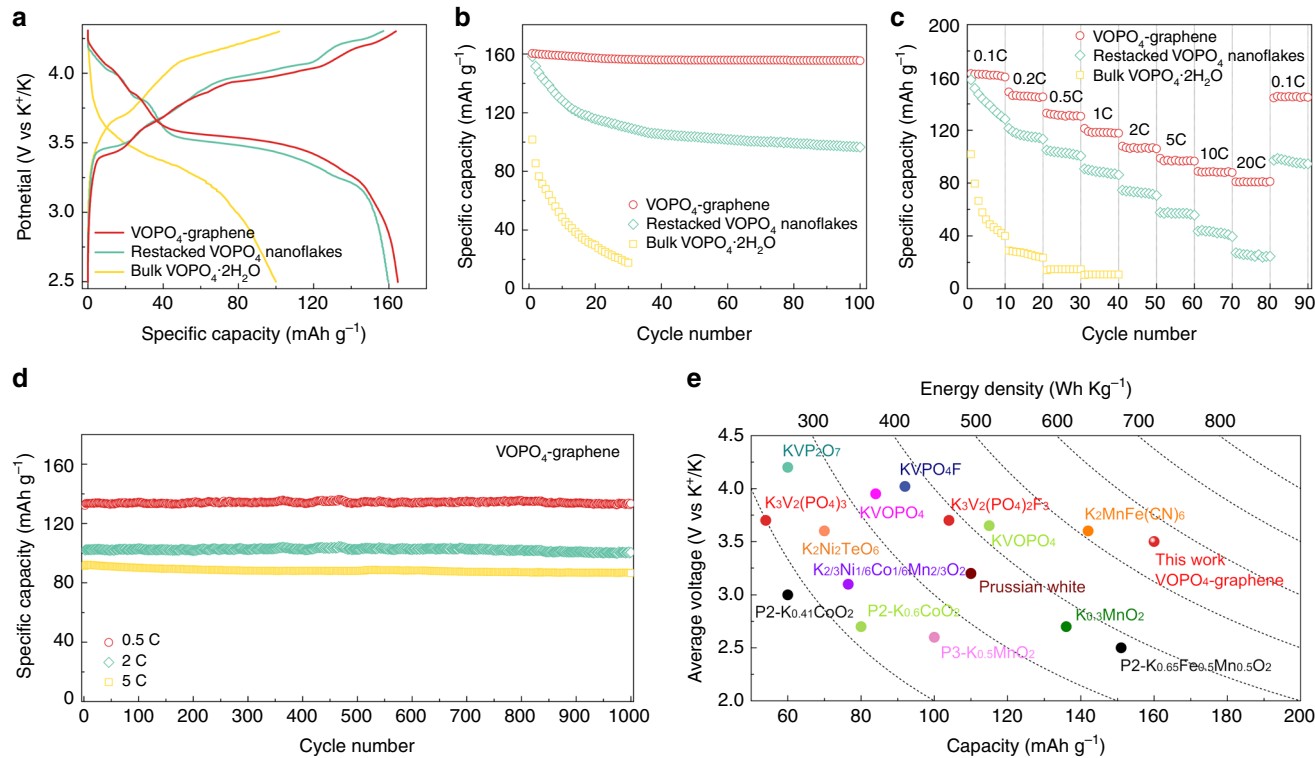

**Fig. 5 PIB performances. a** Charge/discharge profiles of bulk VOPO$_4$·2H$_2$O, restacked VOPO$_4$ nanoflakes and VOPO$_4$-graphene at 0.1C (16 mA g$^{-1}$). **b** Comparison of cycling performance of bulk VOPO$_4$·2H$_2$O, restacked VOPO$_4$ nanoflakes and VOPO$_4$-graphene at 0.1C. **c** Rate capability of bulk VOPO$_4$·2H$_2$O, restacked VOPO$_4$ nanoflakes and VOPO$_4$-graphene at various current densities. **d** Long-term cycling performance of VOPO$_4$-graphene at current densities of 0.5, 2, and 5C. **e** Comparison of VOPO$_4$-graphene cathode with some other reported cathodes for PIBs. Further details pertaining to the selected cathode materials are provided in the Supplementary Table 1.

demonstrates the long cycle life of the 2D VOPO$_4$-graphene multilayered heterostructure. After 1000 cycles, high reversible capacities of ~135, 105, and 85 mA h g$^{-1}$ were sustained at 0.5, 2, and 5C, respectively. Supplementary Fig. 25a shows the cyclic voltammetry (CV) curves of the VOPO$_4$-graphene cathode in PIBs at various scan rates. Similar shapes and a gradual broadening of redox peaks were observed. The redox peaks in CV scans are consistent with the voltage plateaus in voltage profiles, indicating a high reversibility. To explore the mechanism for the superior electrochemical performance of VOPO$_4$-graphene, we performed kinetics analysis based on CV measurements. The relationship between peak current ($i$) and scan rate ($v$) was investigated, according to $i = av^b$[45]. The value of $b = 0.5$ indicates a diffusion-controlled process, whereas $b = 1$ means a capacitive-dominated process. The $b$-value can be obtained by plotting log($i$) versus log($v$) (Supplementary Fig. 25b). The values of 0.84 and 0.82 were obtained for cathodic and anodic peaks, respectively, indicating a capacitive-dominated process. Further calculation of the diffusion and capacitive contributions at a certain scan rate can be done by separating the specific contribution from the capacitive-controlled ($k_1v$) and diffusion-controlled ($k_2v^{1/2}$) process according to $i(V) = k_1v + k_2v^{1/2}$[45]. At the scan rate of 0.3 mVs$^{-1}$, a capacitive contribution was calculated to be ~81% (Supplementary Fig. 25c). This capacitive-dominated K storage process supports a K$^+$ ion intercalation/deintercalation mechanism without a phase and structural change in the 2D multilayered VOPO$_4$-graphene heterostructure[21,45,46]. This behavior verifies the reasons for the superior rate capability and cycling stability of the VOPO$_4$-graphene cathode.

The potassiation/depotassiation potential of the cathode has a significant effect on the output energy density of full PIBs.

However, most of the previously developed cathodes for PIBs exhibited lower energy density than their analogs for LIBs and SIBs[3]. Figure 5e shows a comparison of the specific capacity, average voltage, and energy density of some typical PIB cathodes, including layered transition metal oxides, polyanionic compounds, and Prussian Blue analogs, etc. Layered transition metal oxides, such as K$_{0.3}$MnO$_2$ and K$_{0.65}$Fe$_{0.5}$Mn$_{0.5}$O$_2$, generally exhibit high reversible capacities but suffer from a low voltage[47–50]. Polyanion cathodes have higher voltages than their oxide counterparts, owing to the enhanced ionic bonds originated from the introduction of polyanion moiety with high electronegativity[51]. However, their specific capacity is generally limited to ~100 mA h g$^{-1}$. The Prussian Blue analogs such as K$_2$MnFe(CN)$_6$ are promising cathode materials due to their high voltage and theoretical capacity[52–54]. Unfortunately, their residual crystal water may block the insertion of K$^+$ ions, and even cause safety issues in 'high' voltage operation. Impressively, the 2D VOPO$_4$-graphene multilayered heterostructure cathode can deliver a high potassiation/depotassiation potential of ~3.5 V, which results in the highest energy density of ~570 W h kg$^{-1}$ among recently reported cathode materials for PIBs (Fig. 5e and Supplementary Table 1).

The electrochemical behavior of 2D VOPO$_4$-graphene multilayered heterostructure was also investigated as cathodes for sodium-ion batteries (SIBs), zinc-ion batteries (ZIBS) and aluminum-ion batteries (AIBs). As shown in Fig. 6a and b, a similar trend of reversible shift of 001 peak during the intercalation/de-intercalation of Na$^+$ ions was observed for 2D VOPO$_4$-graphene multilayered heterostructure. A continuous lattice volume variation during charge/discharge cycling (Fig. 6c) resulted in a small volume change of only 1.7%, suggesting a zero-strain feature of 2D VOPO$_4$-graphene multilayered

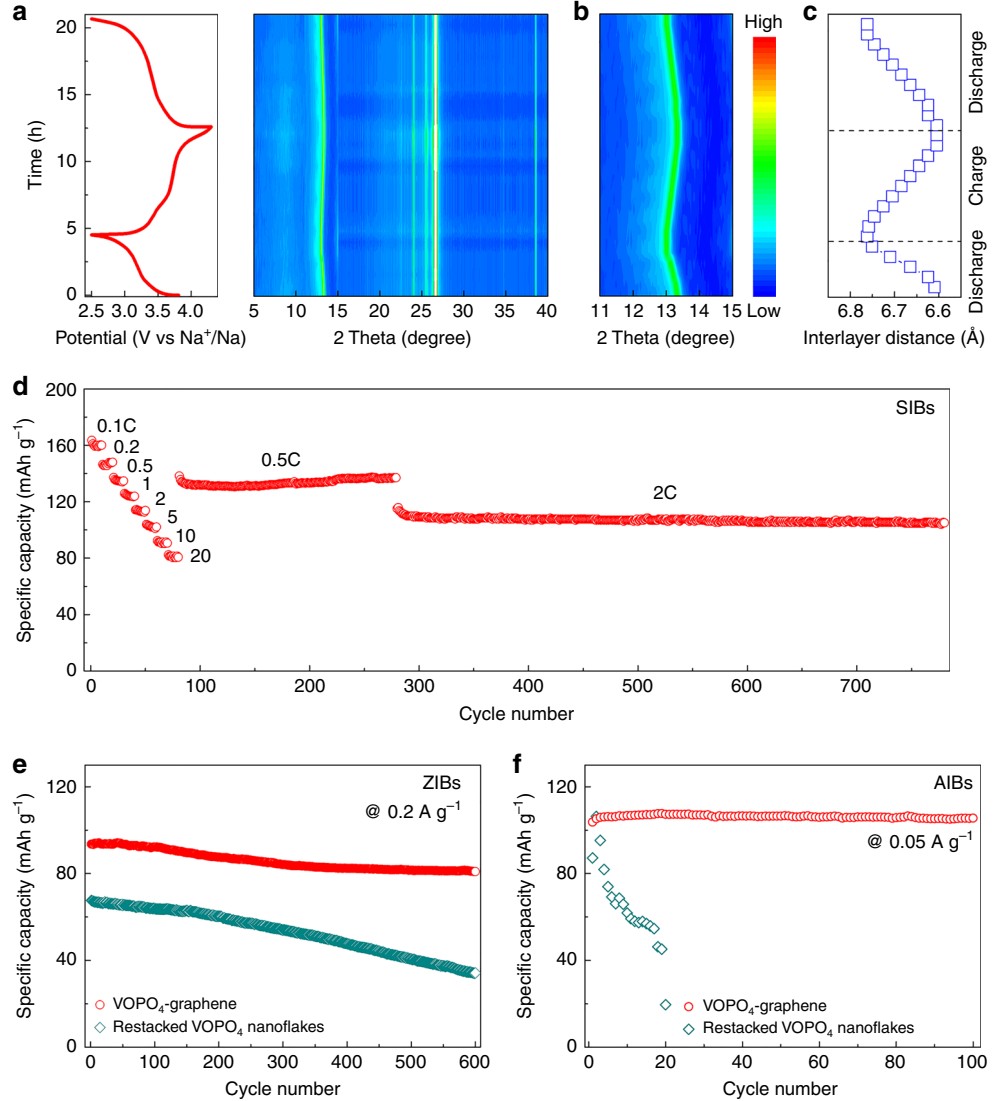

**Fig. 6 SIB, ZIB, and AIB performances. a** Typical charge/discharge profiles and in-situ XRD patterns of $VOPO_4$-graphene as cathodes for SIBs. **b** Enlarged in-situ XRD patterns from 11° to 15° of $VOPO_4$-graphene. **c** The interlayer distance of $VOPO_4$-graphene calculated from XRD patterns during the charge/discharge processes of SIBs. **d** Rate capability and cycling stability of $VOPO_4$-graphene as cathodes for SIBs. **e** Comparison of cycling performance of restacked $VOPO_4$ nanoflakes and $VOPO_4$-graphene as cathodes for ZIBs. **f** Comparison of cycling performance of restacked $VOPO_4$ nanoflakes and $VOPO_4$-graphene as cathodes for AIBs.

heterostructure as intercalation cathodes for SIBs. Consequently, the 2D $VOPO_4$-graphene multilayered heterostructure exhibited superior cycling stability compared with the restacked $VOPO_4$ nanoflakes (Supplementary Fig. 26). When cycled at different current densities, the 2D $VOPO_4$-graphene multilayered heterostructure cathode also exhibited an excellent rate capability and a high cycling stability (Fig. 6d and Supplementary Fig. 27). The CV curves showed a high reversibility of the 2D $VOPO_4$-graphene multilayered heterostructures as cathodes for Na storage (Supplementary Fig. 28). Supplementary Figs. 29 and 30 show the typical charge/discharge profiles of the restacked $VOPO_4$ nanoflakes and 2D $VOPO_4$-graphene multilayered heterostructure as cathodes for ZIBs at different current densities, respectively. The predominantly sloping profiles with the same shape suggest a solid-solution type process of highly reversible intercalation of $Zn^{2+}$, resulting in a high rate capability (Supplementary Fig. 31). Supplementary Fig. 32 shows the CV curves of the 2D $VOPO_4$-graphene multilayered heterostructures as cathodes in ZIBs. The almost rectangular CV curves are in

accordance with the slope shapes in the voltage profiles. Upon continuous cycling at 0.2 A g$^{-1}$, a specific capacity of around 85 mA h g$^{-1}$ was maintained after 600 cycles (Fig. 6e). The 2D $VOPO_4$-graphene multilayered heterostructure cathode also showed promising electrochemical activity for $Al^{3+}$ (Supplementary Figs. 33 and 34). Under 0.05 A g$^{-1}$, almost no capacity decay was observed after 100 cycles (Fig. 6f and Supplementary Fig. 35). In contrast to the intercalation of $AlCl_4^-$ anions in graphite[55], the 2D $VOPO_4$-graphene multilayered heterostructure is prone to insert trivalent $Al^{3+}$ ions as with the reported vanadium-based layered materials[56–58]. Although the polarization should be addressed, the high cycling stability of 2D $VOPO_4$-graphene multilayered heterostructure cathodes provides a new avenue for multivalent ion-based energy storage.

## Discussion
In summary, we have demonstrated that interface strain in a 2D multilayered $VOPO_4$-graphene heterostructure produces a

zero-strain intercalation cathode for beyond Li$^+$-ion batteries such as Na$^+$, K$^+$, Zn$^{2+}$, and Al$^{3+}$ ion batteries. The 2D multi-layered heterostructure cathodes can undergo a reversible inter-calation of both monovalent and multivalent metal ions, with a negligible volume change. Consequently, high-performance K$^+$-ion batteries with a stable reversible capacity of 160 mA h g$^{-1}$ and a high energy density of ~570 W h kg$^{-1}$ have been achieved by employing the 2D multilayered heterostructures as cathodes. Moreover, as cathodes for Na$^+$, Zn$^{2+}$, and Al$^{3+}$ ion batteries, the 2D multilayered heterostructures can also deliver a high cycling stability. The strategy of strain engineering could be extended to many other nanomaterials for rational design of electrode materials towards high energy storage applications beyond lithium-ion chemistry.

## Methods

**Synthesis of bulk layered VOPO$_4$·2H$_2$O crystals**. Bulk layered VOPO$_4$·2H$_2$O crystals were synthesized according to a method reported in the previous literature[30]. Briefly, a mixture of V$_2$O$_5$ (4.8 g), H$_3$PO$_4$ (85% 26.6 mL), and H$_2$O (115.4 mL) was refluxed at 110 °C for 16 h. The resulting yellow precipitate was collected by centrifugation, washed several times with water and acetone, and then dried in oven at 60 °C.

**Synthesis of VOPO$_4$ nanosheets**. The VOPO$_4$ nanosheets were synthesized by an intercalation-exfoliation process[59,60]. For the synthesis of intercalated VOPO$_4$, the bulk VOPO$_4$·2H$_2$O (1 g), acrylamide (10 g) and ethanol (50 mL) were mixed and stirred at 30 °C for 72 h. The product was washed with acetone, and then dried under ambient conditions. The VOPO$_4$–acrylamide intercalation compound was dispersed in isopropanol with a concentration of 2 mg mL$^{-1}$ and then followed by stirring at room temperature for 24 h. The obtained suspensions were allowed to stand for more than 7 days, after which the supernatant was separated and collected for use. In a direct exfoliation process[61], bulk VOPO$_4$·2H$_2$O was dispersed in isopropanol with a concentration of 2 mg mL$^{-1}$ and then ultra-sonicated for 30 min. The suspension was separated and collected for use.

**Synthesis of VOPO$_4$-graphene multilayered heterostructures**. The modified graphene nanosheets with a positively charged nature were synthesized according to our previous studies[20–22]. The VOPO$_4$-graphene multilayered heterostructures were prepared by a solution-phase self-assembly strategy. Specifically, the suspensions of VOPO$_4$ and modified graphene nanosheets were mixed dropwise under continuous stirring. The flocculated VOPO$_4$-graphene was collected and washed by centrifugation and then freeze-dried.

**Material characterization**. Powder XRD data were recorded using a Bruker D8 Discover diffractometer equipped with monochromatic Cu Kα radiation (λ = 0.15405 nm). A field-emission scanning electron microscope (FE-SEM, Zeiss Supra 55VP) and a JEOL JEM-ARM200F TEM instrument were employed to observe the microstructures and morphologies of the samples. A Dimension 3100 SPM instrument was used to examine the topography of the nanosheets deposited onto Si wafer substrates. The zeta potentials of nanosheet suspensions were determined using an ELS-Z zeta-potential analyzer. The thermogravimetric (TG) analysis were carried out using an SDT 2960 thermoanalyzer under an air atmosphere from 25 to 800 °C with a heating rate of 5 °C min$^{-1}$. XPS measurements were performed using an ESCALAB250Xi (Thermo Scientific, UK) equipped with mono-chromated Al K alpha (energy: 1486.68 eV). Raman spectra were obtained from a Renishaw inVia spectrometer system (Gloucestershire, UK) equipped with a Leica DMLB microscope (Wetzlar, Germany) and a Renishaw He–Ne laser source.

**Electrochemical measurements**. All the battery measurements were carried out using a half-cell system in CR2032-type coin cells. The specific capacity was calculated based on the mass ratio of the VOPO$_4$ active materials. The cathodes were prepared by making a slurry of active material, carbon black, and poly(tetra-fluoroethylene) (PTFE) in a weight ratio of 80:15:5. For the PIBs, the slurry was casted onto Al foils with a mass loading of the active material of approximately 1.5 mg cm$^{-2}$. Potassium foils were used as the counter and reference electrodes. Whatman glassy fibers were used as the separators. The electrolyte was 0.8 M KPF$_6$ in ethylene carbonate (EC)/propylene carbonate (PC) (1/1 V/V) with a 5 wt% fluoroethylene carbonate (FEC) additive. For the SIBs, the slurry was casted onto Al foils with a mass loading of the active material of approximately 1.5 mg cm$^{-2}$. Sodium foils were used as the counter and reference electrodes. Whatman glassy fibers were used as the separators. The electrolyte was 1 M NaClO$_4$ in PC with 2% FEC additive. For the ZIBs, the slurry was coated on a stainless steel foil with an active-material mass loading of 1.5 mg cm$^{-2}$. 1 M Zn(CF$_3$SO$_3$)$_2$ in acetonitrile was used as an electrolyte. A zinc foil and a filter paper were used as anode and

separator, respectively. For the AIBs, the slurry was pasted on a Mo foil with an average mass loading of the active material of 1.0 mg cm$^{-2}$. The electrolyte was a room temperature ionic liquid (RTIL) consisting of aluminum chloride and 1-ethyl-3-methylimidazolium chloride (molar ratio AlCl$_3$: [EMIm]Cl = 1.5). Aluminum disk was used as the anode electrode. A glass fiber filter paper was employed as the separator.

**Theoretical calculation**. All the calculations were performed by the framework of spin polarized DFT as implemented in the Vienna ab-initio Simulation Package (VASP)[62,63]. The exchange-correlation potentials were treated by the generalized gradient approximation (GGA) parameterized by Perdew, Burke and Ernzerhof (PBE)[64]. The interaction between valence electrons and ion cores was described by the projected augmented wave (PAW) method[65] and the DFT-D2 method[66–68] was adopted for the multilayered systems taking into account of van der Waals (vdW) interaction. The electronic wave functions were expanded in a plane-wave basis with a cutoff energy of 400 eV. For two-dimensional bilayer or four layered VOPO$_4$, the lattice parameters along two periodic directions were set as $a = b = 6.02$ Å, and for the nonperiodic direction, the lattice character was set as $c = 35$ Å. As for the consideration of the bilayer or multilayer systems with edges, one-dimensional models were constructed with the lattices parameter of $b = 6.02$ Å. Besides, the lattice characters along the other directions were set as $a = 22$ Å and $c = 35$ Å, respectively. The reciprocal space was sampled by $5 \times 5 \times 1$ and $1 \times 5 \times 1$ grid meshes for the geometry optimization of 2D and 1D systems, by using the Monkhorst–Pack scheme[69]. All the atoms were fully relaxed until the Hellmann–Feynman force on each atom was less than 0.01 eV Å$^{-1}$.

## Data availability

All relevant data are available from the corresponding author upon reasonable request.

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

## Acknowledgements

This work was financially supported by the Australian Research Council through the ARC Discovery projects (DP160104340, DP170100436, and DP180102297). This work was supported in part by the WPI-MANA, Ministry of Education, Culture, Sports, Science and Technology, Japan, and CREST of the Japan Science and Technology Agency (JST). X.Z. acknowledge the support of the NSFC (11574262). C.Z. and Z.L. acknowledge the facilities and the technical assistance of the Microscopy Australia Node at the University of Sydney (Sydney Microscopy and Microanalysis).

## Author contributions

P.X., T.S., and G.W. designed the research. P.X. and F.Z. conducted the synthesis, characterizations, and electrochemical measurements. P.X., X.Z., and Y.S. performed the theoretical calculations. S.W., H.L., B.S., and J.Z. helped the electrochemical measurements. C.Z. and Z.L. carried out the microscopy characterizations. P.X., R.M., Y.B., T.S., and G.W. wrote the paper with contributions from other co-authors. All authors contributed to the discussion of the manuscript.

## Competing interests

The authors declare no competing interests.
