## [Peer Review File · Nature Communications]

Reviewers' comments:

Reviewer #1 (Remarks to the Author):

In this manuscript, the authors present a promising strategy for achieving a negligible volume change for the cathode material of VOPO₄. The authors also systematically investigate this material for energy storage, which shows good performance beyond lithium ion batteries. The zero-strain properties of this rationally designed 2D VOPO₄ are further validated by DFT calculations and in-situ XRD. It would be a meaningful contribution in this field. However, some concerns listed below need to be addressed before it can be considered for publishing on Nature Communications.

1. Cyclic voltammetry (CV) data is very important to analyze the electrochemical process upon charge/discharge. The peaks in CV curves can reveal the reversibility of the redox couples involved in the (de)lithiation process. The authors did not provide any CV data in this manuscript to show the reversibility of VOPO₄ and supporting information for each energy storage system.
2. A superior rate performance of VOPO₄-graphene in potassium ion batteries is highlighted in this manuscript. However, based on the details the authors presented (active materials loading: 1.5 mg cm⁻²), the testing current reaches about 2.5 mA at 5C (assuming 16mm electrode in CR2032 coin cell). It has been widely accepted that potassium metal is not a good quasi-reference electrode especially at high current density condition. Thus, to make the C rate and high current density cycling data more convincing, data obtained from a consistent and stable K-K symmetrical cell should provide (at least 5C/10C/20C K-K symmetrical cell data).
3. As the superior rate performance of VOPO₄-graphene has been reported by the authors, it would be better to clarify the capacity contribution mechanism. Considering the nanosheets structure of VOPO₄-graphene and its superior rate performance, a pseudocapacitive capacity contribution is likely existed (iScience, 6, 212-221).

Reviewer #2 (Remarks to the Author):

In this manuscript, the authors synthesized the multilayered VOPO₄-graphene composite materials for K-, Na-, Zn- and Al-ion batteries, and exhibited excellent electrochemical performance. And the authors used various characterization methods along with theoretical calculation. However, I still have some concerns as follows and thus suggest that the manuscript would be accepted after major revision.

1. The as-synthesized VOPO₄-graphene cathode exhibited impressive cycling performance, which is mainly attributed to the multilayered structure. The authors used Cross-section HRTEM image (Figure 2f) to show unique microstructure, which seems to be not convincing. Could the authors provide more solid experimental results?
2. We estimated the theoretical capacity of VOPO₄-graphene cathode for K-ion battery based on the V⁴⁺/V⁵⁺ valence change and the value is 162 mAh/g. In the work, the measured capacity is 160 mAh/g, very close to the theoretical value. That means that in the full charged state the value of element V is +5, which is inconsistent with Figure 3g. In Figure 3g, there exists V⁴⁺ except for V⁵⁺.
3. The as-synthesized VOPO₄-graphene cathode contains no K ion, thus the cell is initially discharged (potassiated). Based on the Figure 2 a and d, the first discharge capacity is lower than the first charge (depotassiated) capacity. Could the authors explain for it?
4. As shown in Figure 3, The 001 peak of as-synthesized 2D VOPO₄-graphene and restacked VOPO₄ nanoflakes shifted to lower 2θ values upon discharge (potassium intercalation) and change reversibly after charge, representing increased interlayer distances when K-ions inserted into the layers and decreased interlayer distances when K-ions extracted. However, for similar layer materials, the interlayer d-spacing would decrease upon discharge (ions intercalation into host materials), owing to an attractive force between the inserted metal ions and the lattice oxygen

atoms of stacked layers. And when ions extracted from host materials, the interlayer d-spacing would change in reverse[1-5]. Could the author explain this change in crystal structure that is different from conventional understanding?

[1] Hyoung J, Heo J W, Chae M S, et al. Electrochemical Exchange Reaction Mechanism and the Role of Additive Water to Stabilize the Structure of $\text{VOPO}_4 \cdot 2 \text{H}_2\text{O}$ as a Cathode Material for Potassium-Ion Batteries. *ChemSusChem*, 2019, 12(5): 1069-1075.

[2] Liao J, Hu Q, Che B, et al. Competing with other polyanionic cathode materials for potassium-ion batteries via fine structure design: new layered KVOPO_4 with a tailored particle morphology. *Journal of Materials Chemistry A*, 2019, 7(25): 15244-15251.

[3] Tian B, Tang W, Su C, et al. Reticular $\text{V}_2\text{O}_5 \cdot 0.6\text{H}_2\text{O}$ Xerogel as Cathode for Rechargeable Potassium Ion Batteries. *ACS Applied Materials & Interfaces*, 2017, 10(1): 642-650.

[4] Wang P, Chen Z, Wang H, et al. A high-performance flexible aqueous Al ion rechargeable battery with long cycle life. *Energy Storage Materials*, 2020, 25: 426-435.

[5] Wang J, Tan S, Xiong F, et al. $\text{VOPO}_4 \cdot 2\text{H}_2\text{O}$ as a new cathode material for rechargeable Ca-ion batteries. *Chemical communications (Cambridge, England)*, 2020.

Response to Reviewers' Comments

We would like to thank all reviewers for taking time and efforts to review our manuscript. We sincerely appreciate all the reviewers for their valuable comments and suggestions, which helped us to improve the overall quality of the manuscript. Our point-by-point responses to the reviewers' comments are described below. All revised contents and added revisions have been added and highlighted in the revised manuscript and revised supporting information.

Reviewer #1:

In this manuscript, the authors present a promising strategy for achieving a negligible volume change for the cathode material of VOPO₄. The authors also systematically investigate this material for energy storage, which shows good performance beyond lithium ion batteries. The zero-strain properties of this rationally designed 2D VOPO₄ are further validated by DFT calculations and in-situ XRD. It would be a meaningful contribution in this field. However, some concerns listed below needs to be addressed before it can be considered for publishing on Nature Communications.

Response: We thank the reviewer for the positive comments on our work. In particular, we appreciate the reviewer's comments that "it would be a meaningful contribution in this field".

Comment 1: *Cyclic voltammetry (CV) data is very important to analyze the electrochemical process upon charge/discharge. The peaks in CV curves can reveal the reversibility of the redox couples involved in the (de)lithiation process. The authors did not provide any CV data in this manuscript to show the reversibility of VOPO₄ and supporting information for each energy storage system.*

Response: We appreciate the reviewer's valuable suggestions. The CV measurements of VOPO₄-graphene as cathodes for sodium-ion batteries (SIBs), potassium-ion batteries (PIBs), zinc-ion batteries (ZIBS) and aluminium-ion batteries (AIBs) have been conducted. The results and related discussions have been added in the revised manuscript.

Corresponding revisions on Page 19 in the revised manuscript:

Figure S25a shows the cyclic voltammetry (CV) curves of VOPO₄-graphene cathode in PIBs at various scan rates. Similar shapes and a gradual broadening of redox peaks were observed. The redox peaks in CV scans are consistent with the voltage plateaus in voltage profiles, indicating a high reversibility.

Corresponding revisions on Page 21 in the revised manuscript:

The CV curves showed a high reversibility of the 2D VOPO₄-graphene multilayered heterostructures as cathodes for Na storage (Figure S28).

Figure S32 shows the CV curves of the 2D VOPO₄-graphene multilayered heterostructures as cathodes in ZIBs. The almost rectangular CV curves are in accordance with the slope shapes in the voltage profiles.

The 2D VOPO₄-graphene multilayered heterostructure cathode also showed promising electrochemical activity for Al³⁺ (Figure S33 and S34).

Corresponding revision in Supporting Information:

Figure S25a. CV curves of VOPO₄-graphene multilayered heterostructures as cathodes for K-ion batteries at various scan rates.

Figure S28. CV curves of VOPO₄-graphene multilayered heterostructures as cathodes for Na-ion batteries at 0.1 mV s⁻¹.

Figure S32. CV curves of VOPO₄-graphene multilayered heterostructures as cathodes for Zn-ion batteries at 0.1 mV s⁻¹.

Figure S34. CV curves of VOPO₄-graphene multilayered heterostructures as cathodes for Al-ion batteries at 0.1 mV s⁻¹.

Comment 2: A superior rate performance of VOPO₄-graphene in potassium ion batteries is highlighted in this manuscript. However, based on the details the authors presented (active materials loading: 1.5 mg cm⁻²), the testing current reaches about 2.5 mA at 5C (assuming 16mm electrode in CR2032 coin cell). It has been widely accepted that potassium metal is not a good quasi-reference electrode especially at high current density condition. Thus, to make the C rate and high current density cycling data more convincing, data obtained from a consistent and stable K-K symmetrical cell should provide (at least 5C/10C/20C K-K symmetrical cell data).

Response: We appreciate the reviewer's valuable suggestion. A symmetric K-K cell was fabricated and cycled at different current densities. As can be seen in Figure S24, the K-K symmetric cell consistently showed stable K stripping/plating polarizations.

Corresponding revisions on Page 18 in the revised manuscript:

A symmetric K-K cell was also fabricated and consistently showed stable K stripping/plating polarizations at different current densities (Figure S24).

Corresponding revision in Supporting Information:

Figure S24. Rate performances of symmetric K-K cells at different current densities.

Comment 3: As the superior rate performance of VOPO₄-graphene has been reported by the authors, it would be better to clarify the capacity contribution mechanism. Considering the nanosheets structure of VOPO₄-graphene and its superior rate performance, a pseudocapacitive capacity contribution is likely existed (iScience, 6, 212-221).

Response: We appreciate the reviewer's valuable suggestions. The kinetics analysis based on the CV measurements have been carried out. The related data and discussion have been added in the revised manuscript.

Corresponding revisions on Page 19 in the revised manuscript:

To explore the mechanism for the superior electrochemical performance of VOPO₄-graphene, we performed kinetics analysis based on CV measurements. The relationship between peak current (i) and scan rate (v) was investigated, according to $i = av^b$ [45]. The value of $b = 0.5$ indicates a diffusion-controlled process, whereas $b = 1$ means a capacitive-dominated process. The b -value can be obtained by plotting $\log(i)$ versus $\log(v)$ (Figure S25b). The values of 0.84 and 0.82 were obtained for cathodic and anodic peaks, respectively, indicating a capacitive-dominated process. Further calculation of the diffusion and capacitive contributions at a certain scan rate can be done by separating the specific contribution from the capacitive- (k_1v) and diffusion-controlled ($k_2v^{1/2}$) process according to $i(V) = k_1v + k_2v^{1/2}$ [45]. At the scan rate of 0.3 mVs^{-1} , a capacitive contribution was calculated to be ~81% (Figure S25c). This capacitive-dominated K storage process supports a K⁺ ion intercalation/deintercalation mechanism without phase and structural change in the 2D multilayered VOPO₄-graphene heterostructure [21,45,46]. This behavior verifies the reasons for the superior rate capability and cycling stability of the VOPO₄-graphene cathode.

Corresponding revision in References:

[21] Xiong, P., Zhang, X., Zhang, F., Yi, D., Zhang, J., Sun, B., Tian, H., Shanmukaraj, D., Rojo, T., Armand, M., Ma, R., Sasaki, T. & Wang, G. Two-dimensional unilamellar cation-deficient metal oxide nanosheet superlattices for high-rate sodium ion energy storage. *ACS Nano* **12**, 12337–12346 (2018).

[45] Augustyn, V., Come, J., Lowe, M. A., Kim, J. W., Taberna, P.-L., Tolbert, S. H., Abruña, H. D., Simon, P. & Dunn, B. High-rate electrochemical energy storage through Li⁺ intercalation pseudocapacitance. *Nat. Mater.* **12**, 518–522 (2013).

[46] Wei, Q., Jiang, Y., Qian, X., Zhang, L., Li, Q., Tan, S., Zhao, K., Yang, W., An, Q., Guo, J. & Mai, L. Sodium ion capacitor using pseudocapacitive layered ferric vanadate nanosheets cathode. *iScience*. **6**, 212–221 (2018).

Corresponding revision in Supporting Information:

Figure S25. (a) CV curves of VOPO₄-graphene multilayered heterostructures as cathodes for K-ion batteries at various scan rates. (b) Determination of the b-value using the relationship between peak current and scan rate. (c) Separation of the capacitive and diffusion currents at a scan rate of 0.3 mV s⁻¹.

Reviewer #2:

In this manuscript, the authors synthesized the multilayered VOPO₄-graphene composite materials for K-, Na-, Zn- and Al-ion batteries, and exhibited excellent electrochemical performance. And the authors used various characterization methods along with theoretical calculation. However, I still have some concerns as follows and thus suggest that the manuscript would be accepted after major revision.

Response: We thank the reviewer for the positive comments on our work. According to the reviewer's suggestions, we had revised our manuscript. The point-by-point responses to the reviewers' comments are described below.

Comment 1: *The as-synthesized VOPO₄-graphene cathode exhibited impressive cycling performance, which is mainly attributed to the multilayered structure. The authors used Cross-section HRTEM image (Figure 2f) to show unique microstructure, which seems to be not convincing. Could the authors provide more solid experimental results?*

Response: We appreciate the reviewer's comments. The VOPO₄-graphene was prepared by a solution-phase flocculation method between the exfoliated VOPO₄ nanosheets and modified graphene layers. The VOPO₄ nanosheets are negatively charged and modified graphene layers are positively charged. After mixing the suspensions of these two nanosheets, a flocculation is formed immediately, which suggests that these nanosheets are restacked together. Due to electrostatic attraction, these two oppositely charged nanosheets are prone to self-assemble on top of each other. Following this behavior, a face-to-face stacked multilayered heterostructure is supposed to be achieved [*J. Am. Chem. Soc.* **129**, 8000–8007 (2007); *J. Am. Chem. Soc.* **137**, 2844–2847 (2015); *ACS Nano* **12**, 1768–1777 (2018); *ACS Nano* **12**, 12337–12346 (2018); *ACS Energy Lett.* **3**, 997–1005 (2018); *Nano Lett.* **19**, 4518–4526 (2019)]. As can be seen in Figure 2f, the HRTEM image of the cross-section reveals that the multilayered structures. The almost amorphous parts should be graphene sheets (marked as G) whereas the crystalline parts with lattice fringes are VOPO₄ sheets (marked as VP). Following to the reviewer's suggestions, we have provided a side-view SEM image (Figure S12), which further confirmed the multilayered heterostructure of VOPO₄-graphene.

Corresponding revision on Page 9 in the revised manuscript:

A side-view SEM image (Figure S12) indicates a multilayered heterostructure of VOPO₄-graphene.

Corresponding revision in Supporting Information:

Figure S12. A side-view SEM image of VOPO₄-graphene showing the multilayered heterostructure morphology.

Comment 2: We estimated the theoretical capacity of VOPO₄-graphene cathode for K-ion battery based on the V⁴⁺/V⁵⁺ valence change and the value is 162 mA h/g. In the work, the measured capacity is 160 mA h/g, very close to the theoretical value. That means that in the full charged state the value of element V is +5, which is inconsistent with Figure 3g. In Figure 3g, there exists V⁴⁺ except for V⁵⁺.

Response: We thank the reviewer's comments. The as-prepared VOPO₄ nanosheets displayed a mixture of +4 and +5 oxidation states [*Chem. Mater.* **14**, 3882–3888 (2002)]. After the 1st discharge, the signal of V⁵⁺ in the XPS spectra was remarkably decreased, confirming the reduction of V during the insertion of K⁺ ions. After a full charge, a major contribution of V⁵⁺ and a small amount of V⁴⁺ were observed, revealing a similar environment to that of the initial samples. This indicated a reversible oxidation and reduction of V^{5+/4+} during the charge and discharge processes. However, the slightly increased quantity of V⁴⁺ state suggested the charged sample is not exactly recovered to the initial state [*Chem. Mater.* **27**, 8211–8219 (2015)]. This should be ascribed to a slightly irreversible process during the 1st charge/discharge process, which is commonly observed in nanostructured electrodes, especially the atomically thin sheet-based heterostructures in this work [*Nano Lett.* **18**,

2402–2410 (2018); *J. Mater. Chem. A* **6**, 15530–15539 (2018); *J. Mater. Chem. A* **2**, 2461–2466 (2014); *Chem. Mater.* **27**, 757–767 (2015); *J. Power Sources* **351**, 35–44 (2017); *Adv. Funct. Mater.* **29**, 1901719 (2019)]. Due to a high specific surface area of the multilayered heterostructure, it is reasonable that the partial of the intercalated K^+ ions are still trapped in the $VOPO_4$ -graphene after the first charge cycle. Correspondingly, we further checked the K 2p signals during the 1st charge/discharge cycle. As shown in Figure S21, the K 2p signals were highly dominated in the discharge process, as evidence for K^+ ion intercalation. However, in the charge process, a slight amount of K was still observed, implying the incomplete extraction of K^+ ions [*ACS Appl. Mater. Interface* **10**, 642–650 (2018); *J. Mater. Chem. A* **7**, 8315–8326 (2019); *ChemSelect* **4**, 11711–11717 (2019)]. For the reviewer's concern about the measured capacity close to the theoretical value, we performed kinetics analysis based on CV measurements to explore the K storage mechanism for the $VOPO_4$ -graphene cathodes. As shown in Figure S25, a capacitive-dominated K storage process was observed. This suggested a reversible intercalation of K^+ ions without phase and structural change in the 2D multilayered $VOPO_4$ -graphene heterostructure [*Nat. Mater.* **12**, 518–522 (2013); *Nat. Commun.* **6**, 6544 (2015); *Small*, **13**, 1702588 (2017); *ACS Nano* **12**, 12337–12346 (2018); *iScience*. **6**, 212–221 (2018)]. Thus, based on the above discussion, it is reasonable to observe the XPS signals of both V^{5+} and V^{4+} in the charged cathode, although the measured capacity is close to the theoretical value.

Corresponding revision on Page 14 in the revised manuscript:

The reversible reduction and oxidation of $V^{5+/4+}$ upon K-ion intercalation and de-intercalation were also observed (Figure 3g). After the 1st discharge, the signal of V^{5+} in the XPS spectra was remarkably decreased, confirming the reduction of V during the insertion of K^+ ions. After a full charge, a major contribution of V^{5+} and a small amount of V^{4+} were observed. However, the slightly increased quantity of V^{4+} state suggested the charged sample is not exactly recovered to the initial state. This should be ascribed to a slightly irreversible process during the 1st charge/discharge process. Due to a high specific surface area of the multilayered heterostructure, it is reasonable that the partial of the intercalated K^+ ions are still trapped in the $VOPO_4$ -graphene after the first charge cycle. Correspondingly, we further checked the K 2p signals during the 1st charge/discharge cycle. As shown in Figure S21, the K 2p signals were highly dominated in the discharge process, as

evidence for K⁺ ion intercalation. However, in the charge process, a slight amount of K was still observed, implying the incomplete extraction of K⁺ ions.

Corresponding revisions on Page 19 in the revised manuscript:

To explore the mechanism for the superior electrochemical performance of VOPO₄-graphene, we performed kinetics analysis based on CV measurements. The relationship between peak current (i) and scan rate (v) was investigated, according to $i = av^b$ [45]. The value of $b = 0.5$ indicates a diffusion-controlled process, whereas $b = 1$ means a capacitive-dominated process. The b -value can be obtained by plotting $\log(i)$ versus $\log(v)$ (Figure S25b). The values of 0.84 and 0.82 were obtained for cathodic and anodic peaks, respectively, indicating a capacitive-dominated process. Further calculation of the diffusion and capacitive contributions at a certain scan rate can be done by separating the specific contribution from the capacitive- (k_1v) and diffusion-controlled ($k_2v^{1/2}$) process according to $i(V) = k_1v + k_2v^{1/2}$ [45]. At the scan rate of 0.3 mVs⁻¹, a capacitive contribution was calculated to be ~81% (Figure S25c). This capacitive-dominated K storage process supports a K⁺ ion intercalation/deintercalation mechanism without phase and structural change in the 2D multilayered VOPO₄-graphene heterostructure [21,45,46]. This behavior verifies the mechanisms for the superior rate capability and cycling stability of the VOPO₄-graphene cathode.

Corresponding revision in References:

[21] Xiong, P., Zhang, X., Zhang, F., Yi, D., Zhang, J., Sun, B., Tian, H., Shanmukaraj, D., Rojo, T., Armand, M., Ma, R., Sasaki, T. & Wang, G. Two-dimensional unilamellar cation-deficient metal oxide nanosheet superlattices for high-rate sodium ion energy storage. *ACS Nano* **12**, 12337–12346 (2018).

[45] Augustyn, V., Come, J., Lowe, M. A., Kim, J. W., Taberna, P.-L., Tolbert, S. H., Abruña, H. D., Simon, P. & Dunn, B. High-rate electrochemical energy storage through Li⁺ intercalation pseudocapacitance. *Nat. Mater.* **12**, 518–522 (2013).

[46] Wei, Q., Jiang, Y., Qian, X., Zhang, L., Li, Q., Tan, S., Zhao, K., Yang, W., An, Q., Guo, J. & Mai, L. Sodium ion capacitor using pseudocapacitive layered ferric vanadate nanosheets cathode. *iScience*. **6**, 212–221 (2018).

Corresponding revision in Supporting Information:

Figure S21. Ex-situ K 2p XPS of VOPO₄-graphene in initial, discharged and charged states.

Figure S25. (a) CV curves of VOPO₄-graphene multilayered heterostructures as cathodes for K-ion batteries at various scan rates. (b) Determination of the b-value using the relationship between peak current and scan rate. (c) Separation of the capacitive and diffusion currents at a scan rate of 0.3 mV s⁻¹.

Comment 3: *The as-synthesized VOPO₄-graphene cathode contains no K ion, thus the cell is initially discharged (potassiated). Based on the Figure 2 a and d, the first discharge capacity is lower than the first charge (depotassiated) capacity. Could the authors explain for it?*

Response: We thank the reviewer's comment. As can be seen in Figure 3a and 3d, a low discharge capacity was obtained during the first charge/discharge cycle. He and Li *et al.* also observed this phenomenon during the first charge/discharge process of VOPO₄ cathode in sodium-ion batteries [*Chem. Mater.* **28**, 682–688 (2016); *Energy Environ. Sci.* **9**, 3399–3405 (2016)]. The possible reason could be the introduction of protons in the structure during the depotassiation process. The related discussion has been added in the revised manuscript.

Corresponding revision on Page 12 in the revised manuscript:

A low discharge capacity was obtained, which possibly originated from the introduction of protons during the depotassiation process [35,36].

Corresponding revision in References:

[35] He, G., Kan, W. H. & Manthiram, A. A 3.4 V layered VOPO₄ cathode for Na-ion batteries. *Chem. Mater.* **28**, 682–688 (2016).

[36] Li, H., Peng, L., Zhu, Y., Chen, D., Zhang, X. & Yu, G. An advanced high-energy sodium ion full battery based on nanostructured Na₂Ti₃O₇/VOPO₄ layered materials. *Energy Environ. Sci.* **9**, 3399–3405 (2016).

Comment 4: *As shown in Figure 3, The 001 peak of as-synthesized 2D VOPO₄-graphene and restacked VOPO₄ nanoflakes shifted to lower 2θ values upon discharge (potassium intercalation) and change reversibly after charge, representing increased interlayer distances when K-ions inserted into the layers and decreased interlayer distances when K-ions extracted. However, for similar layer materials, the interlayer d-spacing would decrease upon discharge (ions intercalation into host materials), owing to an attractive force between the inserted metal ions and the lattice oxygen atoms of stacked layers. And when ions extracted from host materials, the interlayer d-spacing would change in reverse [1-5]. Could the author explain this change in crystal structure that is different from conventional understanding?*

[1] Hyoung J, Heo J W, Chae M S, et al. Electrochemical Exchange Reaction Mechanism and the Role of Additive Water to Stabilize the Structure of $\text{VOPO}_4 \cdot 2\text{H}_2\text{O}$ as a Cathode Material for Potassium-Ion Batteries. *ChemSusChem*, 2019, 12(5): 1069-1075.

[2] Liao J, Hu Q, Che B, et al. Competing with other polyanionic cathode materials for potassium-ion batteries via fine structure design: new layered KVOPO_4 with a tailored particle morphology. *Journal of Materials Chemistry A*, 2019, 7(25): 15244-15251.

[3] Tian B, Tang W, Su C, et al. Reticular $\text{V}_2\text{O}_5 \cdot 0.6\text{H}_2\text{O}$ Xerogel as Cathode for Rechargeable Potassium Ion Batteries. *ACS Applied Materials & Interfaces*, 2017, 10(1): 642-650.

[4] Wang P, Chen Z, Wang H, et al. A high-performance flexible aqueous Al ion rechargeable battery with long cycle life. *Energy Storage Materials*, 2020, 25: 426-435.

[5] Wang J, Tan S, Xiong F, et al. $\text{VOPO}_4 \cdot 2\text{H}_2\text{O}$ as a new cathode material for rechargeable Ca-ion batteries. *Chemical communications (Cambridge, England)*, 2020.

Response: We appreciate the reviewer's comments. We agree with the reviewer that some layered $\text{VOPO}_4 \cdot 2\text{H}_2\text{O}$ [*ChemSusChem* **12**, 1069–1075 (2019); *Energy Storage Mater.* **25**, 426–435 (2020); *Chem. Commun.* **56**, 3805–3808 (2020)] and similar layered KVOPO_4 [*J. Mater. Chem. A* **7**, 15244–15251 (2019)] and $\text{V}_2\text{O}_5 \cdot 0.6\text{H}_2\text{O}$ [*ACS Appl. Mater. Interfaces* **10**, 642–650 (2017)] materials showed decreased *d*-spacing upon ion intercalation. It should be noted that the reported structures were bulk layered compounds with water or metal ions in the interlayer space, which is the same as the bulk $\text{VOPO}_4 \cdot 2\text{H}_2\text{O}$ sample in our work. Due to the intercalated species as pillars, the large interlayer spacing could facilitate easy intercalation and diffusion of metal ions such as K^+ ions in the lattice. Upon discharge, K^+ ions were intercalated whereas the crystal water was extracted simultaneously from the structure. Owing to an attractive force between the inserted K^+ ions and the lattice oxygen atoms of stacked layers, a decreased interlayer spacing was obtained. However, in our work, the materials are two layered VOPO_4 structures based on exfoliated VOPO_4 nanosheets, that are the layered VOPO_4 nanoflakes and VOPO_4 -graphene multilayered heterostructure. They are different from the reported bulk materials. During the exfoliation process of bulk layered $\text{VOPO}_4 \cdot 2\text{H}_2\text{O}$ for the synthesis of VOPO_4 nanosheets, the molecular H_2O was escaped from the interlayers of VOPO_4 [*J. Mater. Chem.* **10**, 737–743 (2000); *J. Mater. Chem.* **11**, 1858–1863 (2001); *Nat. Commun.* **4**, 2431 (2013)]. Consequently, a decreased interlayer spacing was observed for the VOPO_4 nanoflakes and VOPO_4 -graphene. Thus, upon insertion of large K^+ ions, a dominant effect of ion intercalation on the host lattice is an expansion perpendicular to the layers [*MRS Bull.* **12**, 24–28 (1987); *Adv. Sci.* **4**, 1700146 (2017); *Phys. Chem. Chem. Phys.* **19**, 13696–13702 (2017)], which

induces the expansion of interlayer distance of VOPO₄ nanosheets. Besides, it should be noted that the VOPO₄ and graphene nanosheets were held together by electrostatic attraction in the multilayered heterostructure. Moreover, an interface-induced compressive strain on the VOPO₄ layers was formed due to the mismatch of the in-plane lattice spacing between VOPO₄ and graphene. The interface strain between VOPO₄ and graphene could accommodate the lattice expansion originated from the intercalation of K⁺ ions, leading to a stable cycling performance. Based on the above discussion, the different structure change of VOPO₄-graphene should be attributed to the unique multilayered heterostructure and the interface strain between the adjacent VOPO₄ and graphene sheets. The related discussion and references have been added in the revised manuscript.

Corresponding revision on Page 14 - 15 in the revised manuscript:

Unlike the previously reported bulk layered materials [37-41], the 2D VOPO₄-graphene multilayered heterostructure showed increased interlayer spacing upon K⁺ ion intercalation. It should be noted that the reported structures were bulk layered compounds with water or metal ions in the interlayer space. Upon discharge, K⁺ ions were intercalated whereas the crystal water was extracted simultaneously from the structure. Owing to an attractive force between the inserted K⁺ ions and the lattice oxygen atoms of stacked layers, a decreased interlayer spacing was obtained. However, in our work, the 2D VOPO₄-graphene multilayered heterostructure is based on exfoliated VOPO₄ nanosheets, in which no intercalated molecular H₂O is preserved and the interlayer distance is decreased. This 2D multilayered heterostructure is different from the reported bulk materials. Thus, upon insertion of large K⁺ ions, a dominant effect of ion intercalation on the host lattice is an expansion perpendicular to the layers [42,43], which induces the expansion of interlayer distance of VOPO₄ nanosheets. Here, in the 2D multilayered heterostructure, the VOPO₄ and graphene nanosheets were held together by electrostatic attraction. An interface-induced compressive strain on the VOPO₄ layers was formed due to the mismatch of the in-plane lattice spacing between VOPO₄ and graphene. The interface strain between VOPO₄ and graphene is possible to accommodate the expansion for a superior stable intercalation reaction. The reversible change in interlayer distance of VOPO₄-graphene should be attributed to the unique multilayered heterostructure and the interface strain between the adjacent VOPO₄ and graphene sheets.

Corresponding revision in References:

- [37] Hyung, J., Heo, J. W., Chae, M. S. & Hong, S.-T. Electrochemical exchange reaction mechanism and the role of additive water to stabilize the structure of $\text{VOPO}_4 \cdot 2\text{H}_2\text{O}$ as a cathode material for potassium-ion batteries. *ChemSusChem* **12**, 1069–1075 (2019).
- [38] Liao, J., Hu, Q., Che, B., Ding, X., Chen, F. & Chen, C. Competing with other polyanionic cathode materials for potassium-ion batteries via fine structure design: new layered KVOPO_4 with a tailored particle morphology. *J. Mater. Chem. A* **7**, 15244–15251 (2019).
- [39] Tian, B., Tang, W., Su, C. & Li, Y. Reticular $\text{V}_2\text{O}_5 \cdot 0.6\text{H}_2\text{O}$ xerogel as cathode for rechargeable potassium ion batteries. *ACS Appl. Mater. Interfaces* **10**, 642–650 (2017).
- [40] Wang, P., Chen, Z., Wang, H., Ji, Z., Feng, Y., Wang, J., Liu, J., Hu, M., Fei, J., Gan, W. & Huang, Y. A high-performance flexible aqueous Al ion rechargeable battery with long cycle life. *Energy Storage Mater.* **25**, 426–435 (2020).
- [41] Wang, J., Tan, S., Xiong, F., Yu, R., Wu, P., Cui, L. & An, Q. $\text{VOPO}_4 \cdot 2\text{H}_2\text{O}$ as a new cathode material for rechargeable Ca-ion batteries. *Chem. Commun.* **56**, 3805–3808 (2020).
- [42] Dresselhaus, M. S. Intercalation in layered materials. *MRS Bull.* **12**, 24–28 (1987).
- [43] Huang, L., Wei, Q., Xu, X., Shi, C., Liu, X., Zhou, L. & Mai, L. Methyl-functionalized MoS_2 nanosheets with reduced lattice breathing for enhanced pseudocapacitive sodium storage. *Phys. Chem. Chem. Phys.* **19**, 13696–13702 (2017).

REVIEWERS' COMMENTS:

Reviewer #1 (Remarks to the Author):

The author has addressed all of issues raised by this reviewer and it can be published in the current form.

Reviewer #2 (Remarks to the Author):

Thanks for authors' response in detail and the response is reasonable except Comment 3. As the as-synthesized VOPO₄-graphene cathode contains no K ion, the cell will be first discharged (potassiated) and then charged (depotassiated). In the manuscript, the depotassiated capacity is lower than the potassiated capacity, and the authors stated that it possibly originated from the introduction of protons during the depotassiation process. I wonder how to introduce the proton during the depotassiation process and how to lead to the increasing of potassiated capacity.

Response to Reviewers' Comments

Reviewer #1:

The author has addressed all of issues raised by this reviewer and it can be published in the current form.

Response: We appreciate the reviewer's positive feedback.

Reviewer #2:

Thanks for authors' response in detail and the response is reasonable except Comment 3.

As the as-synthesized VOPO₄-graphene cathode contains no K ion, the cell will be first discharged (potassiated) and then charged (depotassiated). In the manuscript, the depotassiated capacity is lower than the potassiated capacity, and the authors stated that it possibly originated from the introduction of protons during the depotassiation process. I wonder how to introduce the proton during the depotassiation process and how to lead to the increasing of potassiated capacity.

Response: We appreciate the reviewer's comments and apologize for not explaining it clearly in our previous response. As the VOPO₄-graphene cathode contains no K⁺ ions, it will be firstly potassiated and then depotassiated [*J. Electrochem. Soc.* **152**, A721–A728 (2005); *Chem. Mater.* **28**, 1503–1512 (2016)]. During the first cycle, a lower discharge capacity was obtained compared to the charge capacity. This suggests more ions could be extracted during the charge cycle. In fact, similar results have also been reported in VOPO₄ cathodes for Li and Na ion batteries [*J. Electrochem. Soc.* **152**, A721–A728 (2005); *Chem. Mater.* **28**, 1503–1512 (2016); *Chem. Mater.* **28**, 682–688 (2016); *Energy Environ. Sci.* **9**, 3399–3405 (2016)]. For example, Arumugam Manthiram *et al.* claimed in their published paper, “*In sodium-ion cells, the first cycle shows a low capacity and large irreversible capacity (Figure S4), which possibly originate from the protons in the structure introduced during the delithiation process*” [*Chem. Mater.* **28**, 682–688 (2016)]. The intercalation chemistry of K⁺ ions in VOPO₄ should be analogous to the Li⁺ and Na⁺ ion counterparts. Thus, we made an analogous assumption that the abnormal behavior of our VOPO₄-graphene cathode is probably attributed to the introduction of protons during the depotassiation process [*Chem. Mater.* **28**, 682–688 (2016); *Energy Environ. Sci.* **9**, 3399–3405 (2016)]. Another possible reason could be related to the structure of VOPO₄. The bulk VOPO₄·2H₂O crystal has two distinct water molecules

[*Aust. J. Chem.* **34**, 2035–2038 (1981)]. One is crystal water in the interlayer, which is removed during the exfoliation process [*Nat. Commun.* **4**, 2431 (2013); *J. Nanopart. Res.* **12**, 417–427 (2010)]. The other one is structural water that is bonded to vanadium atom, which is more strongly bonded to the lattice than the crystal water [*J. Am. Chem. Soc.* **124**, 10157–10162 (2002); *ChemSusChem* **12**, 1069–1075 (2019)]. Some structural water molecules were still resided in the lattice after exfoliation [*ChemSusChem* **12**, 1069–1075 (2019)]. As we have demonstrated that some portions of V^{5+} have been reduced to V^{4+} during the exfoliation process. It is possible that the residual structural water was changed into hydrated proton (H_3O^+) to compensate the total charge balance. The already existed protons thus reduce the amount of K^+ ions that could be inserted into the lattice during the first discharge process, resulting in a low potassiation capacity. Then, upon the first charge process, both K^+ ions and protons could be extracted out of the structure, which suggests a large depotassiation capacity. After the first charge/discharge cycle, due to the completed extraction of protons, the amount of K^+ ions that can be inserted into the lattice return to normal, leading to the increasing of potassiated capacity from the second cycle [*Chem. Mater.* **28**, 1503–1512 (2016)]. The related discussion has been revised as below. We hope the above discussion could explain the remaining concerns of the reviewer.

Corresponding revision in Manuscript Page 12:

A large irreversible capacity was possibly due to the protons in the structure, as elucidated by the previous studies^{35–37}.

Corresponding revision in References:

35. He, G., Kan, W. H. & Manthiram, A. A 3.4 V layered $VOPO_4$ cathode for Na-ion batteries. *Chem. Mater.* **28**, 682–688 (2016).
36. Li, H. et al. An advanced high-energy sodium ion full battery based on nanostructured $Na_2Ti_3O_7/VOPO_4$ layered materials. *Energy Environ. Sci.* **9**, 3399–3405 (2016).
37. Hyung, J., Heo, J. W., Chae, M. S. & Hong, S.-T. Electrochemical exchange reaction mechanism and the role of additive water to stabilize the structure of $VOPO_4 \cdot 2H_2O$ as a cathode material for potassium-ion batteries. *ChemSusChem* **12**, 1069–1075 (2019).